# Global climate and nutrient controls of photosynthetic capacity

Yunke Peng[1,2,3], Keith J. Bloomfield[4], Lucas A. Cernusak [5], Tomas F. Domingues[6] & I. Colin Prentice [4,7,8]✉

There is huge uncertainty about how global exchanges of carbon between the atmosphere and land will respond to continuing environmental change. A better representation of photosynthetic capacity is required for Earth System models to simulate carbon assimilation reliably. Here we use a global leaf-trait dataset to test whether photosynthetic capacity is quantitatively predictable from climate, based on optimality principles; and to explore how this prediction is modified by soil properties, including indices of nitrogen and phosphorus availability, measured in situ. The maximum rate of carboxylation standardized to 25 °C ($V_{cmax25}$) was found to be proportional to growing-season irradiance, and to increase—as predicted—towards both colder and drier climates. Individual species' departures from predicted $V_{cmax25}$ covaried with area-based leaf nitrogen ($N_{area}$) but community-mean $V_{cmax25}$ was unrelated to $N_{area}$, which in turn was unrelated to the soil C:N ratio. In contrast, leaves with low area-based phosphorus ($P_{area}$) had low $V_{cmax25}$ (both between and within communities), and $P_{area}$ increased with total soil P. These findings do not support the assumption, adopted in some ecosystem and Earth System models, that leaf-level photosynthetic capacity depends on soil N supply. They do, however, support a previously-noted relationship between photosynthesis and soil P supply.

[1] Masters Programme in Ecosystems and Environmental Change, Department of Life Sciences, Imperial College London, Ascot, UK. [2] Department of Environmental Systems Science, ETH, Zurich, Switzerland. [3] Swiss Federal Institute for Forest, Snow and Landscape Research (WSL), Birmensdorf, Switzerland. [4] Department of Life Sciences, Imperial College London, Ascot, UK. [5] Centre for Tropical Environmental Sustainability Studies, James Cook University, Cairns, QLD, Australia. [6] FFCLRP, Department of Biology, University of São Paulo, Ribeirão Preto, Brazil. [7] Department of Biological Sciences, Macquarie University, North Ryde, NSW, Australia. [8] Department of Earth System Science, Tsinghua University, Beijing, China. ✉email: c. prentice@imperial.ac.uk

Accurate representation of photosynthetic capacity is critical for modelling the response of terrestrial ecosystems to environmental change[1,2]. Earth System models use the FvCB biochemical model[3] to simulate responses of $C_3$ photosynthesis to environment. The modelled instantaneous carbon-assimilation rate is limited either by $V_{cmax}$ ($\mu mol\ m^{-2}\ s^{-1}$), the maximum rate of carboxylation, or $J$, the light-dependent electron transport rate, which is asymptotic at high light towards $J_{max}$ ($\mu mol\ m^{-2}\ s^{-1}$). Both assimilation rates depend on temperature and on the intercellular partial pressure of $CO_2$ ($C_i$).

Application of the FvCB model[3] requires knowledge of three 'plant-determined' quantities: $V_{cmax}$, $J_{max}$ and the ratio of $C_i$ to the ambient partial pressure of $CO_2$ ($C_a$). This ratio, here called $\chi$, is regulated by stomata. $J_{max}$ and $V_{cmax}$ are closely coordinated[4,5]. More data are available on $V_{cmax}$ because it can be inferred from the light-saturated photosynthetic rate, which is commonly measured in the field[6]. Global models have to contend with the large observed variation (in time and space, and within and between species) of $V_{cmax}$. Data analyses have explored its relationship to leaf nutrients[7–9] and environmental variables[10,11]. Until recently, however, most models have assigned constant values of $V_{cmax}$ at standard temperature (conventionally 25 °C: thus $V_{cmax25}$) for each of a small number of plant functional types (PFTs), and allowed the temperature-dependent values to follow standard (instantaneous) equations of enzyme kinetics. Models also have to represent the plant-type and environmental dependencies of $\chi$ (ref. [12]). Most models assign constant per-PFT values of parameters in one of the two widely used models for the response of stomatal conductance to vapour pressure deficit ($D$). However, these simplifications are not the best possible. $V_{cmax25}$ and $\chi$ commonly vary at least as much within as between PFTs; while $\chi$ has predicted (and observed) relationships to growth temperature ($T_g$) and to elevation above sea level ($z$) through its effect on atmospheric pressure, which are neglected in the standard models[10].

One strand of recent research has accordingly focused on a search for universal responses to environment, applicable to all ($C_3$) plants. Eco-evolutionary optimality hypotheses[12–15] have been invoked in recent efforts to derive general principles for the prediction of plant traits and productivity[10,11,16–18]. The least-cost hypothesis[12,19] proposes that investments in transpiration capacity (maintaining the water transport pathway) and $V_{cmax}$ are balanced so that photosynthesis is achieved at the lowest total cost in maintenance respiration of leaves and stems. Within this framework, $\chi$ varies over a limited range, consistent with tight regulation of the balance between water loss and carbon gain[12]. The hypothesis predicts that $\chi$ should decline with increasing $D$, decreasing $T_g$ and increasing $z$. Each of these predictions is quantitatively supported by global compilations of $\chi$ values inferred from stable carbon isotope measurements in leaves[10,20,21] and wood[22]. The coordination hypothesis provides a framework to predict $V_{cmax}$ from physical environmental variables: irradiance (photosynthetic photon flux density, PPFD) and temperature and $CO_2$ (ref. [23]). The 'strong form'[24] of this hypothesis states that carboxylation and electron transport are co-limiting under typical daytime growth conditions, so that neither is in excess. $V_{cmax25}$ is observed to increase with PPFD, $D$ and $z$ (refs. [10,11,21]), and to decline with $T_g$ (refs. [24,25]). The coordination hypothesis predicts all these observations. The decline with $T_g$ is predicted because less Rubisco (the key carboxylation enzyme) is required to support photosynthesis in warmer environments[24]. The increases with $D$ and $z$ are predicted because greater photosynthetic capacity is required to support a given rate of carbon assimilation at lower $\chi$ (ref. [26]).

Positive relationships between photosynthetic capacities and leaf N ($N_{area}$)[27,28] and leaf P ($P_{area}$)[29–32] are also widely observed. Much leaf N is invested in Rubisco[33–36]. Leaf P is required inter alia for cell membranes, nucleic acid synthesis and for ATP and NADPH production[9,37]. The predictive power of relationships to $N_{area}$ or $P_{area}$ is often weak[11,38–40]; however, recent studies[8,9] have proposed a framework in which $V_{cmax25}$ is constrained by the lesser of two functions, one related to $N_{area}$ and the other to $P_{area}$. Leaf nutrient levels, in turn, may or may not reflect their availability in the soil. $N_{area}$ can be related to soil pH (or fertility) but is not unambiguously related to soil N availability[14], while $P_{area}$ is related to both soil fertility and total soil P[14,41].

Thus, there are two conflicting paradigms to explain worldwide variation in photosynthetic capacity. One emphasizes its predictability from climate, based on optimality principles. The other emphasizes its predictability from leaf nutrients. This second approach has been extended to embrace the assumption that leaf nutrients reflect soil nutrient availability—although this is not universally true[42].

To help resolve this contradiction, we assembled a large global dataset of $V_{cmax25}$, $N_{area}$ and $P_{area}$ data from multiple species and sites. In situ soil measurements (pH, C:N ratio and total P) were available at a subset of the sites. Rather than total soil N, which mainly relates to soil organic content, we used soil C:N as an inverse measure of N availability[43]. We hypothesized that

(1) Photosynthetic capacity is subject to first-order control by climate, as predicted by the coordination and least-cost hypotheses. $V_{cmax25}$ increases in proportion to PPFD and increases towards colder and drier environments, due to greater biochemical investment required when $\chi$ is low.

(2) Photosynthetic capacity is reduced, compared to climate-based predictions, under conditions of low nutrient (N and/or P) availability.

## Results

Theoretically predicted values (see 'Methods') of the derivatives of ln $V_{cmax25}$ against ln PPFD, $T_g$ and ln $D$ are given in Table 1, for comparison with values fitted by statistical models (Table 1, Fig. 1). The value of 1 for the derivative of ln $V_{cmax25}$ with respect to ln PPFD implies proportionality, i.e. a 10% increase in PPFD induces a 10% increase in $V_{cmax25}$. The value of $-0.05\ K^{-1}$ for the derivative of ln $V_{cmax25}$ with respect to $T_g$ implies that a 1 °C

**Table 1 Summary statistics for the climatic dependencies of $V_{cmax25}$ ($\mu mol\ m^{-2}\ s^{-1}$).**

| Predictor for $V_{cmax25}$ | Theoretical value | All-species coefficient $R^2 = 0.17$ | Site-mean coefficient $R^2 = 0.31$ |
|---|---|---|---|
| ln PPFD | 1 | $0.99 \pm 0.22$ | $1.02 \pm 0.21$ |
| $T_g$ | $-0.05\ K^{-1}$ | $-0.04 \pm 0.01\ K^{-1}$ | $-0.04 \pm 0.01\ K^{-1}$ |
| ln $D$ | 0.07 | $0.13 \pm 0.06$ | $0.13 \pm 0.06$ |

Log-transformed photosynthetic capacities standardized to 25 °C were derived for all species and as site means. Theoretical values were obtained by evaluating partial derivatives of Eq. (3) with respect to each variable at the median climate of the global dataset (PPFD = 400 $\mu mol\ m^{-2}\ s^{-1}$, $T_g$ = 25 °C, $D$ = 0.60 kPa). All-species coefficients represent the partial effects of each variable, estimated in a mixed effects model with site and species as random effects. Site-mean coefficients represent the partial effect of each variable, estimated in a fixed effects model. All fitted values are given ±1 standard error.

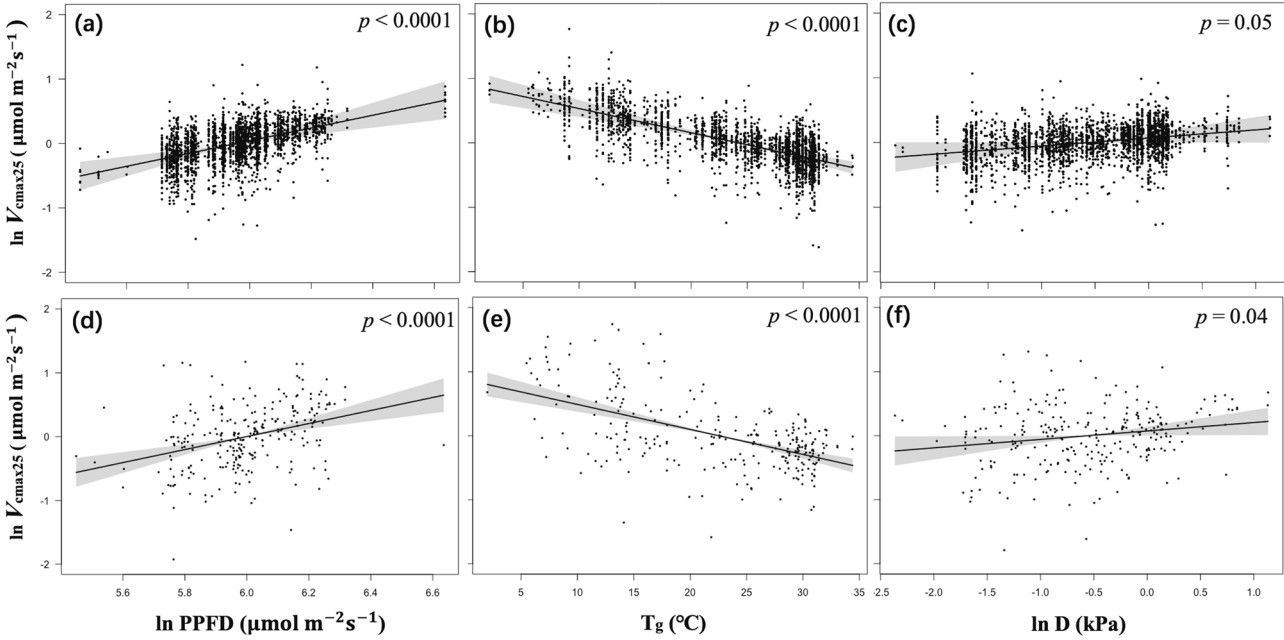

**Fig. 1 Partial residual plots for $V_{cmax25}$ in relation to climate variables.** Partial residual plots for log-transformed $V_{cmax25}$: all-species (**a**, **b**, **c**) and site-means (**d**, **e**, **f**). Coefficients and standard errors for the fitted lines are given in Supporting Information Table S4.

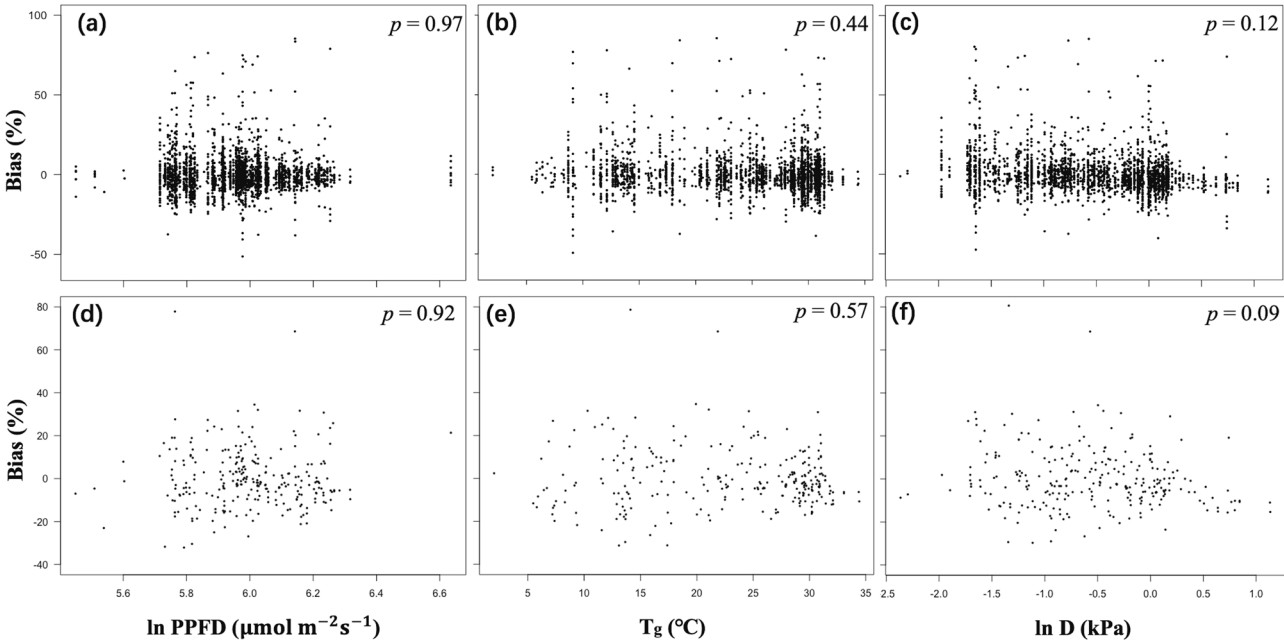

**Fig. 2 Partial residual plots for the model bias of theoretically predicted $V_{cmax25}$ values in relation to climate variables.** Partial residual plots for the model bias of theoretically predicted $V_{cmax25}$ values in relation to climate variables: all-species (**a**, **b**, **c**) and site means (d, e, f). Coefficients and standard errors for the fitted lines are given in Supporting Information Table S4.

increase in growth temperature is predicted to induce a 5% decrease in $V_{cmax25}$. Regression coefficients of $V_{cmax25}$ against the same climate variables were statistically indistinguishable from theoretically predicted values (Table 1). Analysis of site-mean data explained more variance than a mixed-model analysis of all species (see 'Methods'), indicating that a greater fraction of variation in photosynthetic capacity can be explained by physical environmental constraints when considering the whole community together, excluding variation within the community. The response of $V_{cmax25}$ to $D$ was slightly steeper in the 'observed' than the 'theoretical' relationship, but the difference was within

one standard error. From the random term of the all-species mixed model (see 'Methods'), species and site identity separately accounted for 22 and 50% of the variation in $V_{cmax25}$ that was unexplained by the model's climate variables (Table S1).

No significant bias was shown for the predicted relationship of $V_{cmax25}$ to PPFD, $T_g$ or $D$ (Fig. 2). There was a possible under-estimation of $V_{cmax25}$ at higher $D$, but this trend was not sig-nificant either in all-species (Fig. 2c; $p = 0.12$) or site-mean (Fig. 2f; $p = 0.09$) analyses.

Statistical models of photosynthetic capacity (all species and site means) as a function of climate overestimated $V_{cmax25}$ in low-

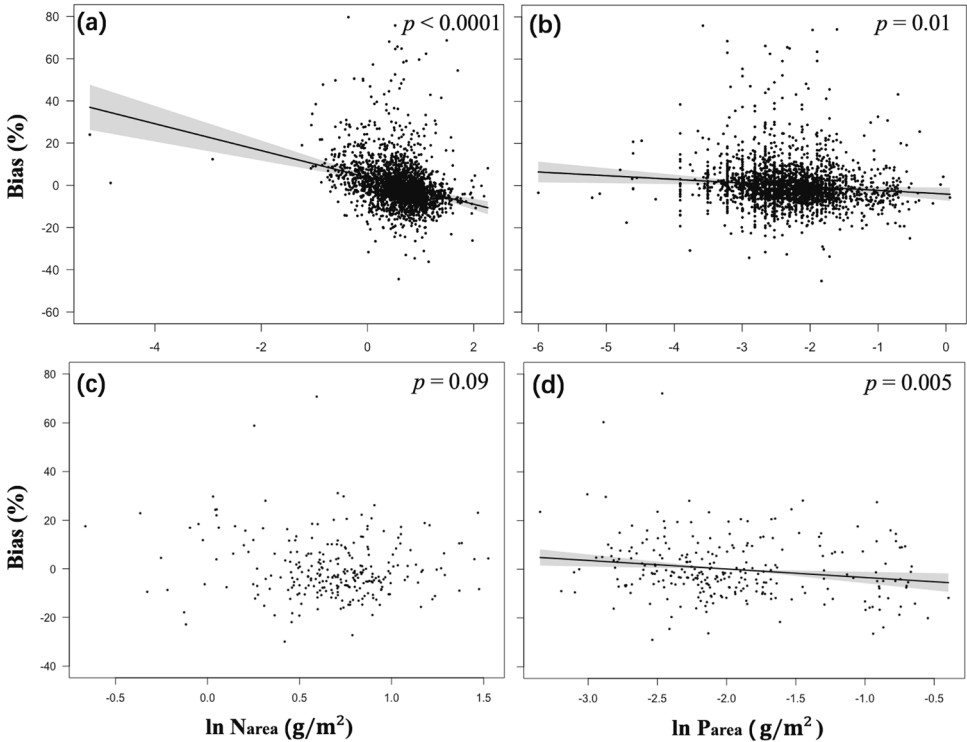

**Fig. 3 Partial residual plots for the model bias of statistically fitted $V_{cmax25}$ in relation to leaf nutrients.** Partial residual plots for the model bias of statistically fitted $V_{cmax25}$ (Table 1) in relation to leaf nutrients, for all-species (**a**, **b**) and site-mean (**c**, **d**) data. The model bias represents the difference between predicted and observed $V_{cmax25}$, where the predicted $V_{cmax25}$ was based on the climate-driven regressions fitted from site-mean and all-species data as shown in Table 1. Coefficients and standard errors for the fitted lines are given in Supporting Information Table S4.

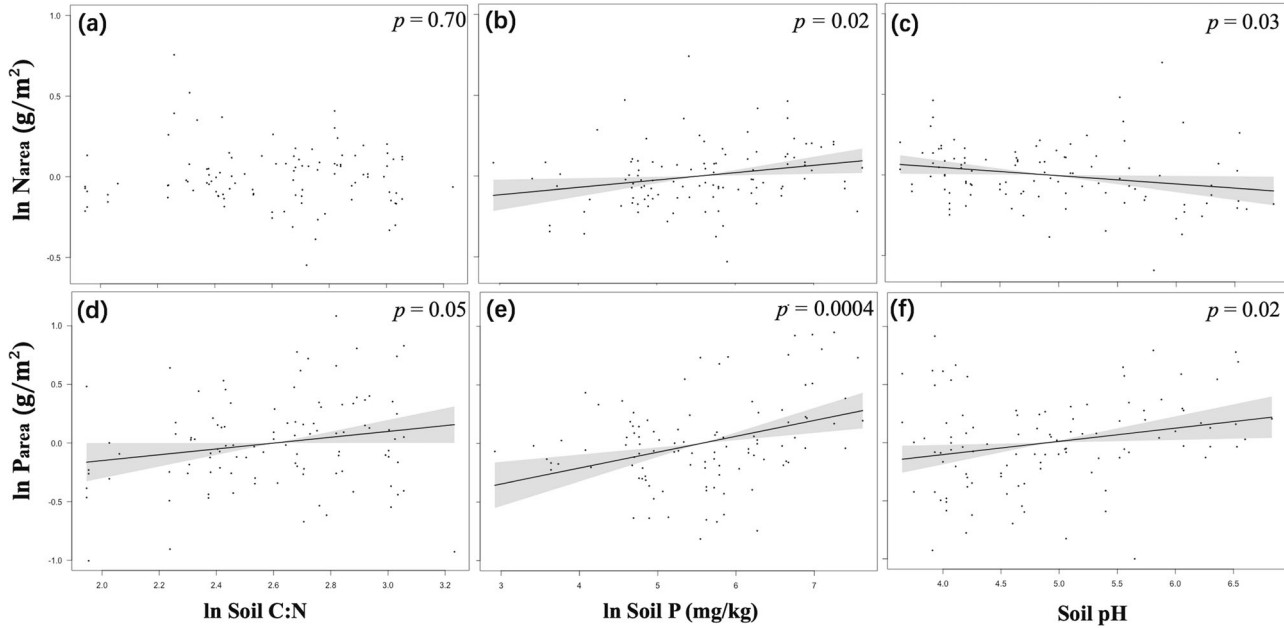

**Fig. 4 Partial residual plots for leaf nutrients in relation to in situ measured soil properties.** Partial residual plots for leaf nutrients (site means) in relation to in situ measured soil properties, for Narea (**a**, **b**, **c**) and Parea (**d**, **e**, **f**) data. Analyses for all species are shown in Fig. S1. Coefficients and standard errors for the fitted lines are given in Supporting Information Table S4.

P leaves and underestimated $V_{cmax25}$ in high-P leaves (Fig. 3b, d). The all-species statistical model also showed a bias in $V_{cmax25}$ related to leaf N (Fig. 3a). This relationship was still apparent ($p < 0.0001$) after removal of three highly influential points. The three

species with extremely low $N_{area}$ values (*Turpinia pomifera*, *Uncaria laevigata* and *Walsura pinnata*) shown in Fig. 3a were sampled in Yunnan, China (21.6°N, 101.5°E). These species possessed very low $V_{cmax25}$ (21 µmol m$^{-2}$ s$^{-1}$) values, probably a

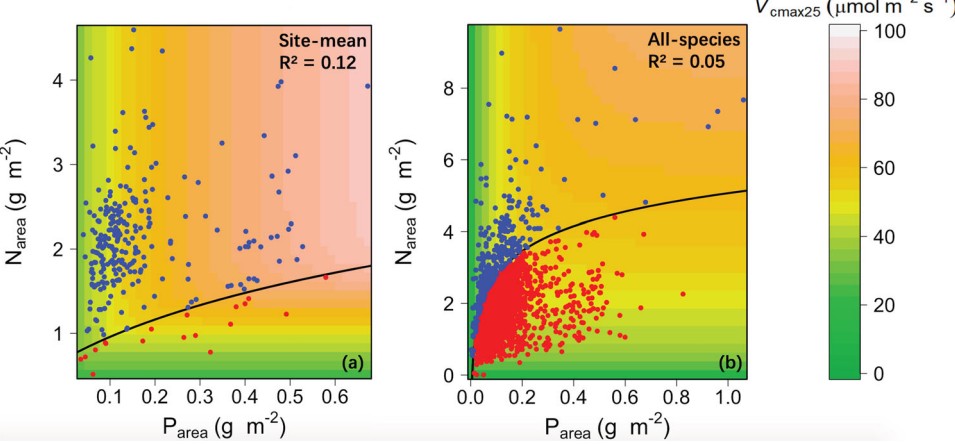

**Fig. 5 Visualizing the co-limitation of $V_{cmax25}$ by Narea and Parea based on the minimum function model.** Visualizing the co-limitation of $V_{cmax25}$ by Narea and Parea for global (**a**) site-mean and (**b**) all-species analyses, based on the minimum function model. Following Domingues et al.8, blue points represent cases where Parea was the 'limiting' nutrient; red points represent cases where Narea was the 'limiting' nutrient. The fitted regression line in (**a**) is Narea = ln (5.96 Parea + 2.01) and in (**b**) is Narea = ln (158.62 Parea + 0.11).

consequence of growth in deep shade. In contrast to the all-species model, the site-mean model showed no bias with respect to $N_{area}$ (Fig. 3c).

Analysis of the subset of the data with in situ soil measurements indicated that $P_{area}$ increased with soil C:N ratio, total soil P and soil pH (Figs. 4d–f, S1d–f). $N_{area}$ increased with soil P (Figs. 4b, S1b) and decreased with soil pH (Figs. 4c, S1c). No relationship was found between leaf N and soil C:N ratio (Figs. 4a, S1a).

Global $V_{cmax25}$ could, alternatively, be represented by a minimum function (Eq. 12) of $N_{area}$ and $P_{area}$. This function provided a better fit to the data than linear regression models with $V_{cmax25}$ as a function of $N_{area}$ and $P_{area}$ and combinations thereof or a model including $N_{area}$, $P_{area}$ and their interaction (Table S2). In site-mean analysis based on the minimum function $P_{area}$ was shown to be the principal limiting factor (93% of sites). In all-species analysis, $N_{area}$ was shown to be the principal limiting factor (86% of species; Fig. 5). This contrast agrees with our findings for model bias: $V_{cmax25}$ variations within sites are more related to leaf N, while variations between sites (community means) are related to mean leaf P but not to mean leaf N. However, the goodness of fit of these models based on nutrients alone ($R^2 = 0.05$, 0.12 for all species and site mean, respectively) was inferior to that of models based on climate alone ($R^2 = 0.17$, 0.31).

## Discussion

The optimality framework accounts for the major global patterns of photosynthetic capacity as shown in our dataset. Consistent with hypothesis (1), global patterns of $V_{cmax25}$ were found to be predictable to first order from PPFD, growth temperature and vapour pressure deficit. Proportionality to PPFD is consistent with observations on light gradients[44], seasonal dynamics[45] and the cloud immersion effect, which decreases PPFD and $V_{cmax25}$ at mid elevations of tropical mountains[39]. $V_{cmax25}$ was predicted (and found) to be greater in drier environments: consistent with the larger biochemical investment required to achieve optimal photosynthesis when stomata are more closed. We found a somewhat steeper than predicted response to $D$ and thus a slight but non-significant underestimation of $V_{cmax25}$ at higher $D$. This might be because the least-cost hypothesis does not consider the compounding effect of low soil moisture, which often accompanies high $D$ and further decreases stomatal conductance, therefore preventing excessive transpiration but increasing the

investment in carboxylation capacity[22,46,47]. In short-term drying experiments, $V_{cmax25}$ typically declines steeply (at different critical pre-dawn water potential values dependent on species[48,49]), although an increase in leaf-level $V_{cmax25}$—which may be accompanied by a reduction in leaf area—can be observed when plants are allowed to acclimate to moderate drought[50–55]. These findings are consistent with the expectation[56] that a decrease of $V_{cmax}$ under drought conditions is linked to a declining hydraulic capacity of the soil–root–xylem system, which can be accommodated over time by leaf shedding. $V_{cmax25}$ showed a negative response to growth temperature, which is predicted because greater investment in photosynthetic enzymes is required at lower temperatures to produce the same catalytic activity[10,55]. Thermal acclimation according to this optimality principle is supported by evidence for a decline of light use efficiency[57] and an increase of photosynthetic nitrogen use efficiency[58] towards warmer environments, and by increased $V_{cmax25}$ at higher elevations[21,41]. The percentage variance explained by these relationships is modest, however (31% for site-mean data: Table 1), consistent with findings by van der Plas et al.[59] on the limits to predictability of ecosystem function from plant traits.

Our hypothesis (2) is partially supported by the analysis of bias in the statistically fitted model. Consistent with findings by Maire et al.[14], we showed an overestimation of $V_{cmax25}$ in leaves with low $P_{area}$. These are typical of sites on acid soils and/or low soil P availability, including some wet tropical forests[14,39]. Many tropical soils are characterized by low total soil P due to long-term weathering[60,61], and a dependency of net primary production on P availability has been shown in tropical forests[62]. Small-scale experimental studies have also suggested that low soil P availability can decrease the light-saturated photosynthetic rate ($A_{sat}$)[63–65] and $V_{cmax}$[66,67]. Adaptation strategies to cope with long-term P deficiency include restricting export of triose phosphate to the cytosol[68], preventing the phosphorylation of ADP to ATP[37,69], phosphate recycling during photorespiration[70] and the replacement of phospholipids by galactolipids and sulpholipids[71,72], all potentially entailing additional costs to the plant. On the other hand, photosynthesis in tropical forests is typically not limited by N[73].

The global relationship between $V_{cmax25}$ and $N_{area}$[74] primarily reflects the large amount of N invested in Rubisco and other photosynthetic enzymes[75]. Leaves with a high photosynthetic capacity necessarily have a large N content per unit area. Within vegetation canopies, $V_{cmax25}$ and $N_{area}$ both vary greatly,

especially along the light gradient from the canopy top to the understory—as shown in many empirical studies[23,35,76,77] and further discussed elsewhere[78,79]. Our data provide no information on the range of light environments within sites and, therefore, our finding of a bias (low-N leaves having lower than predicted $V_{cmax25}$) in the all-species analysis is no surprise. However, the relationship disappeared in the site-mean analysis, indicating that $V_{cmax25}$ at community level is predictable without the need to consider leaf N. Moreover, we found no support for the hypothesis (assumed in some ecosystem and Earth System models) that leaf N is determined by soil N availability—suggesting that the metabolic component of leaf N is determined by photosynthetic capacity, as proposed by Dong et al.[28], rather than vice versa. We did however find that leaf N increases with soil P, which is consistent with the observed effect of soil P on photosynthetic capacity.

A limitation of our analysis is its implicit assumption that mesophyll conductance ($g_m$) is not limiting to photosynthesis. $V_{cmax}$ as estimated here, therefore, is an 'apparent' value and likely to underestimate the true photosynthetic capacity by a variable amount, which cannot be predicted from data currently available at a large scale. However, this simplification reflects the situation in the great majority of ecosystem models, and it has been indicated that 'greater process knowledge of $g_m$ will be required before it can be included [in models]' (ref. [80], p. 26). A more comprehensive understanding of the relationships between leaf nutrients and photosynthesis will depend on advances in understanding the anatomical and physiological controls of $g_m$ (refs. [81,82]), and extensions of leaf-level optimality theory to consider these controls.

In conclusion, while the short-term control of photosynthesis is relatively well understood (and modelled), the longer-term control of photosynthetic capacity is different, and subject to conflicting interpretations. Our findings show that the first-order climatic controls of $V_{cmax25}$ are relatively strong and predictable, indicating that models must account for them. Our results are not consistent with the model assumption that soil N availability controls leaf N, which in turn controls $V_{cmax25}$. They are, however, consistent with previous observational and experimental results indicating the existence of P limitation on leaf P, leaf N and $V_{cmax25}$.

## Methods

During photosynthesis, $C_i$ declines relative to $C_a$ because C assimilation removes $CO_2$ from the intercellular spaces while the stomata impose a resistance to the diffusion of $CO_2$ into the leaf from the air. The $C_i/C_a$ ratio ($\chi$) is maintained within a limited range (about 0.5–0.9 in $C_3$ plants) that is determined by the growth environment[83]. According to the least-cost hypothesis[12,19], $\chi$ is controlled by stomata in such a way as to minimize the sum of the unit costs of the required capacities for transpiration and carboxylation. A consequence of this hypothesis is that for any given set of environmental conditions, there is an optimal value of $\chi$[10,12]

$$\chi_{opt} = \frac{\Gamma^*}{C_a} + \frac{(1 - \frac{\Gamma^*}{C_a})\xi}{\xi + \sqrt{D}}, \text{ where } \xi = \sqrt{\left[\frac{\beta(K + \Gamma^*)}{1.6 \, \eta^*}\right]} \tag{1}$$

that satisfies the least-cost criterion. Here, $\Gamma^*$ is the photorespiratory compensation point, i.e. the value of $C_i$ at which gross photosynthesis is zero; $K$ is the effective Michaelis–Menten coefficient of Rubisco (Pa); $D$ is the leaf-to-air vapour pressure deficit (Pa); $\eta^*$ is the viscosity of water relative to its value at 25 °C and $\beta$ is the ratio of the unit costs of the required capacities for carboxylation and transpiration activities at 25 °C, estimated as 146 based on a global compilation of leaf stable carbon isotope measurements[10]. $K$ is given by

$$K = K_C(1 + O/K_O) \tag{2}$$

where $K_C$ and $K_O$ are the Michaelis–Menten coefficients of Rubisco for $CO_2$ and $O_2$, respectively (Pa, reflecting the twin affinities of Rubisco), and $O$ is the partial pressure of $O_2$ (Pa). $\Gamma^*$, $K_C$ and $K_O$ are functions of temperature, which we apply based on in vivo measurements on tobacco plants[84]. $\Gamma^*$, $C_a$ and $O$ also vary with elevation, in direct proportion to atmospheric pressure.

The coordination hypothesis states that under typical daytime growth conditions photosynthesis is co-limited by carboxylation and electron transport. Optimal $V_{cmax}$ is calculated as

$$V_{cmax, opt} = \varphi_0 I_{abs}[(C_i + K)/(C_i + 2\Gamma^*)] \tag{3}$$

where $\varphi_0$ is the intrinsic quantum efficiency of photosynthesis (mol C mol$^{-1}$ photons); $I_{abs}$ is the PPFD absorbed by the leaf (μmol photons m$^{-2}$ s$^{-1}$). These values were corrected to 25°C using the Arrhenius equation with activation energies from Bernacchi et al.[84,85]. Intrinsic quantum efficiency was assumed to follow the temperature dependency of electron transport in light-adapted leaves[85]

$$\varphi_0 = (0.352 + 0.021 \, T_g - 3.4 \times 10^{-4} T_g^2)/8 \tag{4}$$

According to Eq. (3) and its derivatives, optimal $V_{cmax}$ increases in proportion to PPFD. It also increases with $T_g$. On the other hand, optimal $V_{cmax25}$ declines with $T_g$. This is because the enzyme-kinetic effect, leading to a reduced $V_{cmax25}$ requirement at higher temperatures (caused by the temperature dependency of Rubisco activity), is stronger than the photorespiratory effect, leading to an increased $V_{cmax}$ requirement at higher temperatures (caused by the temperature dependencies of $K$ and $\Gamma^*$). Experimental manipulations of growth temperature[86], repeated measurements on the same plants at different seasons[24], global spatial patterns of $V_{cmax}$[11] and variations of $V_{cmax25}$ on a long elevation transect[41] are all consistent with the negative temperature dependency of $V_{cmax25}$ implied by Eq. (3).

Quantitative predictions of the effect of each climate variable on ln $V_{cmax25}$ can be obtained by taking partial derivatives of Eq. (3) with respect to each variable in turn[21]. Logarithmic transformation is appropriate for magnitude variables described by multiplicative expressions like these[87]. The theory predicts approximately linear relationships of ln $V_{cmax25}$ to ln PPFD, ln $D$ and (without transformation) $T_g$[21]. These derivatives were evaluated at the median climate of the dataset (PPFD = 400 μmol m$^{-2}$ s$^{-1}$, $T_g$ = 25°C, $D$ = 0.60 kPa) using the deriv package in R (ref. [88]) (Table 1).

**Photosynthetic data**. The leaf-trait dataset comprised measurements at 266 sites for a total of 1637 species and 5000 individuals, and soil measurements for 39% of sites (Fig. S2). The dataset consists of field measurements made in natural (unfertilized) vegetation, from several published data sources[7,8,14,20,28,73,89–94]. The numbers of species recorded within each PFT (ref. [95]) are provided in Table S3. $V_{cmax}$ values were derived either from $CO_2$ response ($A$–$C_i$) curves (94% of the dataset) or the one-point method[6] from single measurements of light-saturated net photosynthesis ($A_{sat}$) (6% of the dataset). The one-point method provides a way to estimate $V_{cmax}$ knowing only $A_{sat}$, day respiration ($R_d$), temperature and atmospheric pressure

$$V_{cmax}[est] \approx (A_{sat} + R_d)(C_i + K)/(C_i - \Gamma^*). \tag{5}$$

If no respiration measurement was available, the following approximation was used instead

$$V_{cmax}[est] \approx A_{sat}/[(C_i - \Gamma^*)/(C_i + K) - 0.015] \tag{6}$$

where $R_d$ is assumed to be 1.5% of $V_{cmax}$[6,40,96]. Rogers et al.[97] indicated that the one-point method could result in a twofold underestimation of photosynthetic capacity in the Arctic region. Burnett et al.[98] however estimated errors in photosynthetic capacity at around 20% at most, suggesting that $V_{cmax}$ data obtained in this way (which, in any case, constitute only a small fraction of the dataset) can be justified in the context of a global survey. If measurements were made at a temperature other than 25 °C, reported $V_{cmax}$ and $J_{max}$ values were standardized to 25° C using activation energies provided by Bernacchi et al.[84,85].

**Climate data**. Monthly average values of mean daily maximum ($T_{max}$, °C) and minimum ($T_{min}$, °C) temperatures were extracted at the 0.5° grid location of each site from Climate Research Unit data (CRU TS 4.01)[99], either for the measurement year or for the period 1991–2010 at sites not reporting measurement year. These data were three-dimensionally interpolated to actual site locations (longitude, latitude, elevation) using Geographically Weighted Regression (GWR) in ArcGIS. Mean daytime air temperature ($T_g$) was estimated for each month by assuming the diurnal temperature cycle to follow a sine curve, with daylight hours determined by latitude and month

$$T_g = T_{max}\left\{1/2 + (1 - x^2)^{1/2}/2 \, \cos^{-1} x\right\} + T_{min}\left\{1/2 - (1 - x^2)^{1/2}/2 \, \cos^{-1} x\right\}, \, x = -\tan\lambda\tan\delta \tag{7}$$

where $\lambda$ is latitude and $\delta$ is the monthly average solar declination[100]. Monthly values of $T_g$ were averaged over the thermal growing season, i.e. months with mean daily temperature > 0 °C.

Incident solar radiation data were derived from WATCH Forcing Data ERA-Interim[101] at the same period and resolution, and also interpolated by GWR. Solar radiation (W m$^{-2}$) was converted to PPFD by multiplication by the energy-to-flux conversion factor 2.04 (μmol J$^{-1}$)[102]. PPFD was averaged across the thermal growing season. Mean atmospheric pressures ($P_{atm}$) were derived using the barometric formula[102,103]. $D$ (kPa) was estimated using the Magnus–Tetens

formula[46]

$$D = e_s - e_a, \tag{8}$$

with

$$e_s = 0.611 \ \exp \left[17.27 \ T/(T+237.3)\right], \ \text{where} \ T = (T_{\min} + T_{\max})/2 \tag{9}$$

and

$$e_a = [P_{atm} W_{air} R_v]/[R_d + W_{air} R_v] \tag{10}$$

where $W_{air}$ is the mass mixing ratio of water vapour to dry air; $W_{air} = q_{air} / (1 - q_{air})$, where $q_{air}$ is the specific humidity (kg/kg) derived from WATCH Forcing Data ERA-Interim[101], $R_d$ and $R_v$ are the specific gas constants of dry air and water vapour, $R_d = R/M_d$ and $R_v = R/M_v$, where $R$ is the universal gas constant (8.314 J$^{-1}$ K$^{-1}$), $M_d$ is the molar mass of dry air (28.963 g mol$^{-1}$) and $M_v$ is the molar mass of water vapour (18.02 g mol$^{-1}$).

**Statistical analysis**. The climate data were used to make theoretical predictions of relationships between photosynthetic capacity and climate variables based on the optimality framework, and independently, to derive statistical relationships by multiple regression (Tables S2 and S4). Separate statistical analyses were carried out for individual species, and for site-averaged measurements. In the analyses of individual species (i), each data-point represents the average of one or more measurements on a particular species at a site ($n = 2513$). In the analyses of site-averaged measurements (ii), each data-point represents an average for a site (across all individual and species; $n = 266$) (Table 1). Analyses of type (i) ('all species') data were carried out by means of a linear mixed effects model using the nlme package in R[88]. Climate variables ($T_g$, $D$, PPFD) were included as fixed terms, with site and species as random intercepts. A crossed rather than a fully nested random design was used because some species occurred at more than one site. Ordinary least squares multiple linear regression, using the lm function in R[88], was used for analyses of type (ii) ('site mean') data. Regression relationships were visualized using partial residual plots, obtained with the visreg package in R[88]. Partial residual plots display the relationship between values of the response variable versus each predictor variable, after those responses have been adjusted to hold all other predictors constant at their median values in the dataset. Photosynthetic capacities, PPFD and $D$ were natural log-transformed before analysis so that the resulting regression coefficients can be directly compared with theoretical predictions (Table 1).

**Model data comparisons**. Model bias ($B$, %) in $V_{cmax25}$ was calculated as follows:

$$B = 100\left(V_{cmax25}\left[\text{pred}\right] - V_{cmax25}[\text{obs}]\right)/V_{cmax25}[\text{obs}] \tag{11}$$

where $V_{cmax25}[\text{pred}]$ is a predicted value and $V_{cmax25}[\text{obs}]$ an observed value. Using theoretically predicted values, we explored whether $B$ was significantly related to the climate variables. If so, this would indicate that the true responses of $V_{cmax25}$ to climate variables were different from the predicted ones—pointing to something missing (or wrong) in the theory. Then, we explored whether bias in the values predicted by the statistical models (both all-species and site-mean models) was significantly related to leaf $N_{area}$ and $P_{area}$. If found, such bias would indicate effects of leaf nutrients, additional to the effects of the climate variables considered.

**Alternative models for the response to leaf nutrients**. An alternative statistical model for photosynthetic capacity is a 'minimum function' of $N_{area}$ or $P_{area}$[8]. The following differentiable equation is almost exactly equivalent to a minimum function (Fig. S3):

$$Z = -\left(1/k\right) \ \ln \left[e^{-kx} + e^{-ky}\right] \tag{12}$$

where $Z$ is the response variable ($V_{cmax25}$), $x$ and $y$ are the predictor variables ($N_{area}$, $P_{area}$) and $k \gg 1$. Equation (12) is the 'log-sum-exp' formula, which provides a continuous approximation to the minimum function—allowing its use in regression, and comparison of goodness-of-fit statistics with ordinary linear regression (Table S2). The larger the value of $k$, the closer the approximation to the minimum function. A simple sensitivity analysis showed that large values of $k$ ($\geq 10$) gave best performance (Table S5), indicating that the minimum function fitted the data better than a smooth transition between N and P limitation. Equation (12) was fitted to both all-species and site-mean data (Fig. 5). The equation was plotted using an iterative least squares procedure using the akima, stats and grDevices packages in R[88].

**Statistics and reproducibility**. Data collection, formulae and statistical analyses are described in 'Methods'. All statistical analyses used R software (ref. [88]), applying ordinary linear regression for site-mean analysis and a mixed effects model for all-species analysis. All R packages applied are referenced in 'Methods'. The relevant statistics for the main analyses are presented in Supplementary Information.

**Reporting summary**. Further information on research design is available in the Nature Research Reporting Summary linked to this article.

**Data availability**
No new data were collected for this analysis. The photosynthesis, leaf-trait and soils data are available from the authors of papers cited in the 'Methods' section[7,8,14,20,28,73,89–94]. The complete photosynthesis, climate, leaf-trait and soils datasets underlying all analyses are also publicly available at Zenodo[104] and GitHub: https://github.com/yunkepeng/VcmaxMS. In case of any issues concerning the observed and predicted data and for all queries on ancillary information including the climate data, please contact Y.P. (yunke.peng@usys.ethz.ch) or C.P. (c.prentice@imperial.ac.uk).

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

## Acknowledgements
The authors thank Professor Jon Lloyd for help with obtaining data and many scientific discussions. The authors also thank Tiina Tosens for a helpful review of an earlier draft. This research has received funding from the European Research Council (ERC) under the European Union's Horizon 2020 research and innovation programme (grant agreement no: 787203 REALM) and it is a contribution to the Imperial College initiative on Grand Challenges in Ecosystems and the Environment. The authors thank Owen Atkin, Vincent Maire, Anthony Walker, Han Wang and Huiying Xu for contributing published and/or unpublished data for these analyses. Yanzheng Yang and Yuhui Wu assisted with data collection in China, during fieldwork supported by the National Natural Science Foundation of China (grant agreement nos: 91837312, 31971495). Some data used were collected as part of the UK Natural Environment Research Council consortium 'Tropical Biomes In Transition' (TROBIT) (grant agreement no: NE/D01185x/1) to the University of Edinburgh. Further data were collected within the Nordeste project, supported in Brazil by FAPESP (grant agreement no: 2015-50488-5) and in the UK by NERC and the Newton Fund (grant agreement no: NE/N01256/1).

## Author contributions
I.C.P. proposed the topic and supervised the research. Y.P. carried out all analyses, created the graphics and wrote the first draft of the manuscript. All authors provided data for the analysis and contributed to the interpretation of results and revisions of the manuscript.

## Competing interests
The authors declare no competing interests.
