## [Peer Review File · Communications Biology]

Reviewers' comments:

Reviewer #1 (Remarks to the Author):

Peng and colleagues present a study about the predictability of V_{cmax} and photosynthetic capacity from climatic factors and nutrient availability. The greatest impetus for modern ecophysiological research is the search for universal trait combinations that enable the accurate prediction of photosynthetic parameters in C3 species. Studies based on analyses of global databases are valuable because they facilitate the ways by which the central tendency about the global controls of V_{cmax} and photosynthesis can be determined, just like the approach utilised in the study of Peng and collaborators. One of their novel results shows that contrary to common belief, leaf photosynthetic capacity and V_{max} are not necessarily predictable from the soil nitrogen supply and area-based leaf nitrogen content. Overall, the study of Peng and colleagues is at maximum actuality. They have chosen the appropriate methods to analyse the data, and the final conclusions drawn from the analyses are appropriate and beneficial for a wide audience of natural scientists. My suggestions for the further improvement of the manuscript are as follows:

- In L69–76: In the discussion, I suggest mentioning the potential effect of ignoring the mesophyll conductance on V_{cmax} and photosynthesis estimation. Here are some suggestions for possible context:

'CO₂ diffusion efficiency from the substomatal cavities to the chloroplasts plays an important role in shaping the photosynthetic capacities and leaf resource use efficiency across the Earth's ecosystems by modifying CO₂ concentration in the chloroplasts. Recent studies demonstrate based on quantitative limitation analysis that mesophyll conductance can limit photosynthesis across a range of 16%–80%, thus underscoring that it is often the most limiting factor of photosynthesis. However, the g_m is still generally ignored in large-scale photosynthesis estimations due to lack of knowledge on how best to model it' (Tosens and Laanisto, 2018, *Journal of Experimental Botany*, vol. 69).

Furthermore, the costs associated with proportionally greater nitrogen investments in the cell walls underpin the low g_m and high C_i-C_c , as well as a biased estimation of V_{cmax} (e.g., Niinemets et al. 2009).

- I recommend providing a supplementary figure showing the number of species in each PFT. This would give the reader an overview of the variability of leaf ecological strategies (e.g., about nitrogen investments and resource use strategies) throughout your database.

Reference:

Niinemets Ü, Díaz-Espejo A, Flexas J, Galmés J, Warren CR. Importance of mesophyll diffusion conductance in estimation of plant photosynthesis in the field. *J Exp Bot*. 2009;60(8):2271-2282. doi:10.1093/jxb/erp063

Tiina Tosens

Reviewer #2 (Remarks to the Author):

Peng and others explore the relationship with soil and leaf nutrients photosynthetic capacity. I found the manuscript to be well-written and sensible. The passage on line 80, motivating the analysis, is particularly compelling. In a way, based on the hypotheses, the authors argue that the least-cost hypothesis is adjusted by the coordination hypothesis under nutrient limitation. My largest question involves the justification of using soil C:N in fertilized sites like most agroecosystems when readily-available N will decouple soil C:N from leaf N. Although no relationship with soil C:N and leaf N was found (line 120), which is probably to be expected. The relationships discovered were relatively weak, which to me does not necessarily invalidate the findings but sufficient evidence is presented for the major point that soil N is largely unrelated to $V_{c,max}$, which is an important point for modeling that pushes the science forward. I recommend considering these minor revisions.

χ₁₂ (should be χ (ref. 12)) will be cleared up by copyediting.

58: define z (I'm assuming it's height, but in 68 it's defined as elevation, which can be confused with elevation above sea level. It's really just height from ground or honestly more accurately the difference in distance between the start and end of the water column, which is of course very difficult to define). Perhaps the authors do refer to elevation and in either case I would like to point out this paper by Katul, Oren, and Leuning (2003):
<https://onlinelibrary.wiley.com/doi/full/10.1046/j.1365-3040.2003.00965.x>

Note also line 183 in response to the above point. $V_{c,max}$ should be related to whole-plant hydraulics, not just D .

93: how were $\ln(D)$ values when D approached 0 treated?

106-7: there is some inconsistency in the consistent use of abbreviations that were defined. See also e.g. line 170 and elsewhere.

130: the R^2 values bring to mind emerging literature that plant traits tend to explain ecological function rather poorly, at least on longer time scales (<https://www.nature.com/articles/s41599-020-01316-9>)

Equation 12 is a clever use of an approximation to a minimum function that can be differentiated.

Reviewers' comments:

Reviewer #1 (Remarks to the Author):

Peng and colleagues present a study about the predictability of V_{cmax} and photosynthetic capacity from climatic factors and nutrient availability. The greatest impetus for modern ecophysiological research is the search for universal trait combinations that enable the accurate prediction of photosynthetic parameters in C3 species. Studies based on analyses of global databases are valuable because they facilitate the ways by which the central tendency about the global controls of V_{cmax} and photosynthesis can be determined, just like the approach utilised in the study of Peng and collaborators. One of their novel results shows that contrary to common belief, leaf photosynthetic capacity and V_{max} are not necessarily predictable from the soil nitrogen supply and area-based leaf nitrogen content. Overall, the study of Peng and colleagues is at maximum actuality. They have chosen the appropriate methods to analyse the data, and the final conclusions drawn from the analyses are appropriate and beneficial for a wide audience of natural scientists.

We appreciate this very positive comment.

My suggestions for the further improvement of the manuscript are as follows:

- In L69–76: In the discussion, I suggest mentioning the potential effect of ignoring the mesophyll conductance on V_{cmax} and photosynthesis estimation. Here are some suggestions for possible context:

‘CO₂ diffusion efficiency from the substomatal cavities to the chloroplasts plays an important role in shaping the photosynthetic capacities and leaf resource use efficiency across the Earth’s ecosystems by modifying CO₂ concentration in the chloroplasts. Recent studies demonstrate based on quantitative limitation analysis that mesophyll conductance can limit photosynthesis across a range of 16%–80%, thus underscoring that it is often the most limiting factor of photosynthesis. However, the g_m is still generally ignored in large-scale photosynthesis estimations due to lack of knowledge on how best to model it’ (Tosens and Laanisto, 2018, Journal of Experimental Botany, vol. 69).

Furthermore, the costs associated with proportionally greater nitrogen investments in the cell walls underpin the low g_m and high C_i-C_c , as well as a biased estimation of V_{cmax} (e.g., Niinemets et al. 2009).

We have added some text on this topic. See lines 225–233:

“A limitation of our analysis is its implicit assumption that mesophyll conductance (g_m) is not limiting to photosynthesis. V_{cmax} as estimated here, therefore, is an “apparent” value and likely to underestimate the true photosynthetic capacity by a variable amount, which cannot be predicted from data currently available at a large scale. However, this simplification reflects the situation in the great majority of ecosystem models, and it has been indicated that “greater process knowledge of g_m will be required before it can be included [in models]” (ref.⁸⁰, p. 26). A more comprehensive understanding of the relationships between leaf nutrients and photosynthesis will depend on advances in understanding the anatomical and physiological controls of g_m (refs.^{81,82}), and extensions of leaf-level optimality theory to consider these controls.”

- 80 Rogers, A. *et al.* A roadmap for improving the representation of photosynthesis in Earth system models. *New Phytol.* **213**, 22-42, (2017).
- 81 Tosens, T. & Laanisto, L. Mesophyll conductance and accurate photosynthetic carbon gain calculations. *J. Exp. Bot.* **69**, 5315-5318, (2018).
- 82 Niinemets, Ü., Díaz-Espejo, A., Flexas, J., Galmés, J. & Warren, C. R. Importance of mesophyll diffusion conductance in estimation of plant photosynthesis in the field. *J. Exp. Bot.* **60**, 2271-2282, (2009).

I recommend providing a supplementary figure showing the number of species in each PFT. This would give the reader an overview of the variability of leaf ecological strategies (e.g., about nitrogen investments and resource use strategies) throughout your database.

We now provide this information in Table S3 and refer to it in lines 288:

“The numbers of species recorded within each PFT (ref.⁹⁴) are provided in Table S3.”

- 94 Kattge, J. *et al.* TRY—a global database of plant traits. *Global Change Biol.* **17**, 2905-2935, (2011).

Tiina Tosens

Reviewer #2 (Remarks to the Author):

Peng and others explore the relationship with soil and leaf nutrients photosynthetic capacity. I found the manuscript to be well-written and sensible. The passage on line 80, motivating the analysis, is particularly compelling. In a way, based on the hypotheses, the authors argue that the least-cost hypothesis is adjusted by the coordination hypothesis under nutrient limitation. My largest question involves the justification of using soil C:N in fertilized sites like most agroecosystems when readily-available N will decouple soil C:N from leaf N. Although no relationship with soil C:N and leaf N was found (line 120), which is probably to be expected. The relationships discovered were relatively weak, which to me does not necessarily invalidate the findings but sufficient evidence is presented for the major point that soil N is largely unrelated to $V_{c,max}$, which is an important point for modeling that pushes the science forward. I recommend considering these minor revisions.

The reviewer appreciates our key point that soil N availability, as indexed by soil C:N ratio, is largely unrelated to $V_{c,max}$. N fertilization (decoupling soil C:N from N availability) is not a problem for our analysis, however, because all of our data are from natural (unfertilized) ecosystems. We have updated this in Method Section (lines 286-287):

“The dataset consists of field measurements made in natural (unfertilized) vegetation, from several published data sources.”

χ_{12} (should be χ (ref. 12)) will be cleared up by copyediting.

We have corrected this (line 48).

58: define z (I'm assuming it's height, but in 68 it's defined as elevation, which can be confused with elevation above sea level. It's really just height from ground or honestly more accurately the difference in distance between the start and end of the water column, which is of course very difficult to define). Perhaps the authors do refer to elevation and in either case I would like to point out this paper by Katul, Oren, and Leuning (2003): <https://onlinelibrary.wiley.com/doi/full/10.1046/j.1365-3040.2003.00965.x>

Indeed, z is elevation above sea level, which enters the calculations because of its effect of atmospheric pressure. We have clarified this now (see line 52).

Note also line 183 in response to the above point. $V_{c,max}$ should be related to whole-plant hydraulics, not just D.

We have added this explanation and reference into the text (lines 191-193):

“These findings are consistent with the expectation⁵⁶ that a decrease of $V_{c,max}$ under drought conditions is linked to a declining hydraulic capacity of the soil-root-xylem system, which can be accommodated over time by leaf shedding.”

56 Katul, G., Leuning, R. & Oren, R. Relationship between plant hydraulic and biochemical properties derived from a steady-state coupled water and carbon transport model. *Plant, Cell Environ.* **26**, 339-350, (2003).

93: how were $\ln(D)$ values when D approached 0 treated?

Growing-season average vapour pressure deficit is never zero (in contrast with instantaneous values). The lowest value in our data set is 0.093 kPa, yielding $\ln D = -2.37$, which is not a problematic value for our analysis.

106-7: there is some inconsistency in the consistent use of abbreviations that were defined. See also e.g. line 170 and elsewhere.

We realize that we had not been consistent about the use of abbreviations. We have now edited the MS carefully to ensure that abbreviations are defined at first mention of the quantities they stand for, and used consistently thereafter.

130: the R^2 values bring to mind emerging literature that plant traits tend to explain ecological function rather poorly, at least on longer time scales (<https://www.nature.com/articles/s41559-020-01316-9>)

In light of this comment, we have added the following in the Discussion (lines 198–200).

The percentage variance explained by these relationships is modest, however (31% for site-mean data: Table 1), consistent with findings by van der Plas *et al.*⁵⁹ on the limits to predictability of ecosystem function from plant traits.

59 van der Plas, F. *et al.* Plant traits alone are poor predictors of ecosystem properties and long-term ecosystem functioning. *Nature ecology & evolution* **4**, 1602-1611, (2020).

Equation 12 is a clever use of an approximation to a minimum function that can be differentiated.

We appreciate this comment!